# Estimation of Water Footprint for Major Agricultural and Livestock Products in Korea

**Ik Kim [1],\* and Kyung-shin Kim [2]**

[1]  SMaRT-Eco Consulting Firm, Seoul 06338, Korea
[2]  Department of Environment & Energy Engineering, Sungshin University, Seoul 01133, Korea;
    kyskim@sungshin.ac.kr
\*  Correspondence: kohung@smart-eco.co.kr; Tel.: +82-10-8233-0904

**Abstract:** The Republic of Korea is the only country classified with severe water stress among the 34 Organization for Economic Co-operation and Development (OECD) member countries. Additionally, the self-sufficiency rate of grain in Korea is 27%, which is 1/3 the average of OECD member countries. Because food cannot be produced without water, demand-driven water management of agricultural and livestock products applying water footprints is needed for food security. For this, this study estimates the water footprints of 42 agricultural products and three livestock products. Based on the results, the water footprint of the vegetables grown in facility such as a greenhouse is 7.9 times larger per ton than the footprint of the vegetables cultivated in the open field. Furthermore, the water footprint per ton of beef is about 4.2 times the average water footprint per ton of vegetables grown in facility. Based on the water footprint data of 45 agricultural and livestock products, the footprint of total agricultural and livestock products in 2014 is approximately 27.9% of the total domestic water resources consumed in Korea.

**Keywords:** water footprint; agricultural and livestock products; Penman–Monteith equation; evapotranspiration; climate conditions

## 1. Introduction

The Republic of Korea has experienced rapid industrialization and urbanization since the 1970s. In 2015, the United Nations reported that Korea was the 23rd most densely populated country in the world with 509 people/km$^2$ [1]. Rapid industrialization and urbanization have led to a continual decline in the ground water level. Korea's average annual precipitation is 1274 mm, which is 1.6 times that of the average global precipitation. However, the precipitation per capita is 2660 m$^3$/year, which is 1/6 of the world's average because of the high population density [2]. Kim's study on the Sustainable Water Management Legislation for Climate Change Response published at Yonsei University revealed that the number of extreme drought events in Korea will double in the next 100 years, and the mean drought duration will increase six fold [3]. Additionally, in March 2012, the OECD published their Environmental Outlook to 2050, stating that Korea was classified as the only country with severe water stress among 34-member countries with a stress ratio of over 40% [4]. The ecological footprint is used as an indicator expressing the levels of human consumption. It is a metric of the biologically productive area needed to provide for everything that people use from nature (e.g., fruits and vegetables, fish, wood, fibers, absorption of carbon dioxide from fossil fuel use, and space for buildings and roads). The Global Footprint Network reported in August 2018 that Korea's ecological footprint stood at 8.5 and those of Japan, the UK, the USA, and France were 7.8, 4.0, 2.3, and 1.7, respectively [5]. Thus, the report showed that Korea's consumption was much higher than that of the other OECD member countries.

Foods such as vegetables and meat are among the most important items affecting the ecological footprint. Korea's cereal self-sufficiency rate, including feedstuffs, is 27%, which is 1/3 of the average (83%) of OECD member countries [6,7]. Food production and processing requires a large amount of water. If Korea's food self-sufficiency rate is 100%, there will be a greater demand for water. In fact, Korea imports nearly 3/4 of its food consumed. Therefore, a large amount of water in foreign countries is being used to grow their food. This water called "virtual water" [8]. The importance of virtual water trading has been emphasized to solve the problem of global water depletion [9].

According to the Water Footprint Network (WFN), the concept of water footprint is defined as a measure of humans' appropriation of freshwater in volumes of water consumed and/or polluted [10]. Generally, water is required to manufacture most products. Additionally, water is required to produce agricultural and livestock products. In particular, imported food is produced by using water from the food production site. A food production area can be depleted by exporting large quantities. Thus, water footprint indicators should be used to measure and manage water consumption.

This study estimates the water footprint of 45 agricultural and livestock products in Korea. An extrapolation method is used to extend the water footprints per ton of 45 agricultural and livestock products to the water footprint of the total consumption of 45 products.

## 2. Materials and Methods

### 2.1. Characteristics of Korean Weather

Plants are affected by weather conditions during their growth via evapotranspiration. Weather conditions vary slightly depending on the area. Figure 1 shows a map of South Korea. It has nine Provinces, two special cities, and six metropolitan cities. Korea has a temperate climate with four distinct seasons: spring, summer, autumn, and winter. In spring and autumn, there are many sunny days caused by the migratory anticyclone. In summer, the North Pacific high-pressure system brings hot, humid weather. Winters are cold and dry because of the expanding Siberian high-pressure zone. The average temperature in Korea ranges from 10 to 15 °C with the exception of mountain and island areas. August is the hottest month with temperatures ranging from 23 to 26 °C, whereas the coldest month is January with temperatures between −6 and 3 °C. Annual precipitation per region is 1200–1500 mm in the central region and 1000–1800 mm in the southern region. Gyeongsangbuk Province is ~1000–1300 mm, some parts of Gyeongsangnam Province are about 1800 mm, and Jeju Province is ~1500–1900 mm. Fifty to sixty percent of the annual precipitation falls intensively in summer. Generally, the northwest wind in winter is relatively stronger, and in summer, the southwest wind is stronger [11].

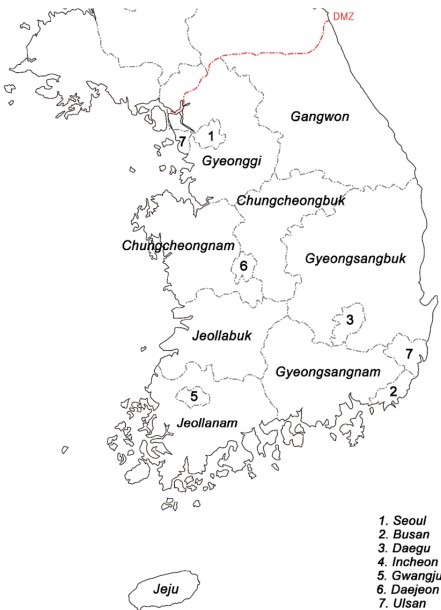

**Figure 1.** Map of provinces of South Korea.

## 2.2. Major Agricultural and Livestock Products Studied

As shown in Table 1, this study selects 45 species that are most commonly consumed by Koreans as the target agricultural and livestock products for estimating the water footprint. The target products combine both the 2014 agricultural and livestock income collection published by the Rural Development Administration and the statistical data on agricultural and livestock products of the National Statistical Office [12,13]. Agricultural products are nine kinds of crops, 13 kinds of open field vegetables, 12 kinds of vegetables in facilities, and eight kinds of fruits. Fruits and vegetables are classified as both open field and in-facility, depending on the cultivation method livestock products such as beef, pork, and chicken are included. Here cultivation in facility means that agricultural products can be grown in agricultural structures such as greenhouses during the winter season.

**Table 1.** Lists of 42 agricultural products and three livestock products.

| Agricultural Products (42) | | | | Livestock Products (3) |
|---|---|---|---|---|
| **Crops (9)** | **Open Field Vegetables (13)** | **Vegetable in Facilities (12)** | **Fruits (8)** | |
| Rice plant<br>Crest<br>Barley<br>Beer barley<br>Corn<br>Bean<br>Sweet potato<br>Spring potato<br>Autumn potato | Spring radish<br>Autumn radish<br>Highland radish<br>Carrot<br>Spring cabbage<br>Autumn cabbage<br>Highland cabbage<br>West cabbage<br>Spinach<br>Watermelon<br>Yellow pepper<br>Garlic<br>Onion | Facility radish<br>Facility cabbage<br>Facility spinach<br>Facility lettuce<br>Facility watermelon<br>Facility melon<br>Cucumber<br>Facility pumpkin<br>Tomato<br>Cherry tomato<br>Strawberry<br>Facility pepper | Apple (open field)<br>Pear (open field)<br>Peach (open field)<br>Persimmon (open field)<br>Grape (open field)<br>Grape (in facility)<br>Tangerine (open field)<br>Tangerine (in facility) | Beef<br>Pork<br>Chicken |

## 2.3. Water Footprint Assessment Model

International Organization for Standardization (ISO) 14046 (2014) and the WFN Water Footprint Assessment Manual are the two methodologies internationally accepted for estimating water footprint in a country, region, or product [10,14]. Both the methodologies present different water types for estimating water footprint. Thus, ISO 14046 classified water types as freshwater, brackish water, surface water, sea water, ground water, and fossil water, whereas the WFN Water Footprint Assessment

Manual classifies water types as green water, blue water, and gray water. The water types of the former are identified according to the water intake point and are closely related to human life and industrial activities. Those of the latter have intimate connections with agricultural activities, which are distinguished by the use of water. It is common to generate and manage operations data in factories by the water types proposed in ISO 14046 in Korea. However, it is more appropriate for the agricultural sector to apply the water types of WFN. Considering the characteristics of the two mentioned methods, the method for estimating the water footprint for the major agricultural and livestock products proposed in this study is based on the methodological procedures and requirements of WFN. The major data categories to be collected, as shown in Table 2, are green water, blue water, and gray water, and the detailed water resource type for blue water is set to follow the requirements of ISO 14046. Additionally, the requirements of ISO 14044 are integrated with WFN's methods of calculating the amount of indirect water and estimating the environmental impacts throughout the entire life cycle of agricultural and livestock products [15,16]. Thus, the water footprint defined in this study integrates direct and indirect water footprints.

**Table 2.** Water types by data categories.

| Activities | Data Category | Water Type | | |
| --- | --- | --- | --- | --- |
| | | **Green** | **Blue** | **Gray** |
| Direct Water | Irrigation water | • Groundwater, surface water | | ○ | |
| | Effective rainfall | • Precipitation | ○ | | |
| | Waste water | • COD, GOD, SS, T-N, T-P | | | |
| Indirect Water | Raw material/energy | • Raw material: strain, seedling, fertilizer, | ○ | ○ | ○ |
| | | • Pesticide, farm materials | | | |
| | | • Energy: electricity, lubricant oil, heavy oil | | | |
| | Waste water | • COD, GOD, SS, T-N, T-P | | | ○ |
| | Solid waste | • Sludge, waste package | | ○ | |

According to the proposed method, this study estimates the water footprints for major agricultural and livestock products presented in Table 1. Furthermore, this study defines functional units, which quantifies the description of performance requirements fulfilled by the product system in different ways for agricultural and livestock products. The water footprint of agricultural products was set up in two ways: a ton basis ($m^3$/ton) and a hectare basis ($m^3$/ha). For livestock products, the functional unit was set to a ton basis. As shown in Figure 2, the system boundary for estimating the water footprint of agricultural products includes seeding, planting, cultivation, and harvesting. That for livestock encompasses feed production, feeding, grazing, slaughtering, and processing.

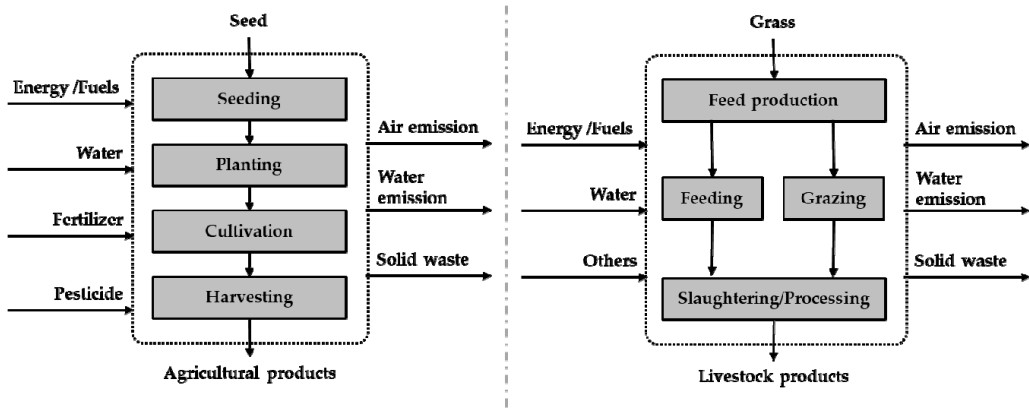

**Figure 2.** System boundaries of agricultural and livestock products.

Table 2 presents data categories collected throughout the life cycle of agricultural and livestock products in 2013. Here, green water includes the on-land precipitation that does not run off or recharge the ground water, but is instead stored in the soil, temporarily staying on top of the soil, or residing in vegetation. Blue water is fresh surface and ground water found in freshwater lakes, rivers, and aquifers [17]. The gray water footprint concept is the amount of fresh water required to assimilate pollutants to meet specific water quality standards. Direct water includes irrigation water (i.e., surface water and groundwater), precipitation, and wastewater, including water quality indicators, such as biochemical oxygen demand (BOD), chemical oxygen demand (COD), suspended solid (SS), total nitrogen (T-N), and total phosphorus (T-P) [18]. The amount of irrigation is calculated by excluding the value of effective rainfall and the cultivation water from the evapotranspiration calculated by the Food and Agriculture Organization (FAO) of the United Nations using the Penman–Monteith equation [17,19–21]. Additionally, because water is not actually wasted in agricultural fields, and agricultural wastewater is not generally found in Korea, this study does not estimate gray water separately. However, gray water estimated from industrial wastewater is considered. Indirect water is calculated by multiplying the collective activities by consumptive water use factors of individual activities. Consumptive water use factors are converted from the national lifecycle inventory database.

### 2.4. Estimation of Direct Irrigation Water

The amount of direct water caused by the evapotranspiration of agricultural products, including crops, is estimated using the FAO Penman–Monteith equation (Figure 3). The technical procedure for measuring direct water quantity for each crop comprises an estimation of evapotranspiration for each crop, a calculation of irrigation water needed, and a measurement of direct water quantity considering scarcity.

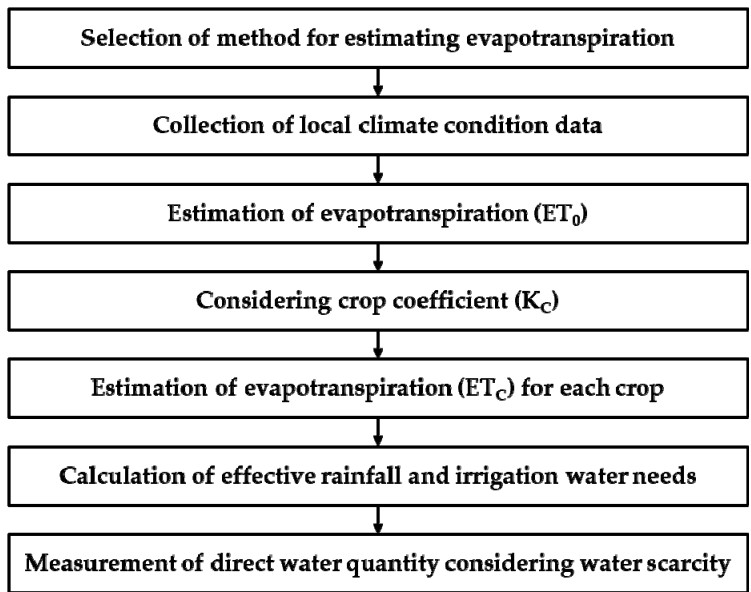

**Figure 3.** Technical procedure for measuring direct water quantity for each crop.

### 2.4.1. Estimation of evapotranspiration for each crop

Evapotranspiration for each crop (*ETc*) is estimated in two steps: net evapotranspiration of the plant (*ETo*) and evapotranspiration for each agricultural product. First, considering Korea's climate,

the Penman–Monteith equation recommended by FAO is used to measure net evapotranspiration. Equation (1) shows the equation [22].

$$ET_0 = \frac{0.408\Delta(R_n - G) + \gamma\frac{900}{T+273}u_2(e_s - e_a)}{\Delta + \gamma(1 + 0.34u_2)} \tag{1}$$

where *ETo* is the reference evapotranspiration (mm/day), Rn is the net radiation at the crop surface (MJ/m$^2$·day), $G$ is the soil heat flux density (MJ/m$^2$·day), $T$ is the mean daily air temperature at 2 m height (°C), $U_2$ is the wind speed at 2 m height (m/s), $e_s$ is the saturation vapor pressure (kPa), $e_a$ is the actual vapor pressure (kPa), $e_s - e_a$ is the saturation vapor pressure deficit (kPa), $\Delta$ is the slope vapor pressure curve (kPa/°C), and $\gamma$ is the psychrometric constant (kPa/°C).

Equation (1) is used to estimate the net evapotranspiration. The following information of daily weather is collected from 79 weather stations in Korea from 2003 to 2012: latitude, longitude, and altitude; maximum and minimum temperature; relative humidity; average vapor pressure; average wind speed; sunshine duration; solar radiation; precipitation; and soil information. Second, the evapotranspiration of each crop was calculated by multiplying the net evapotranspiration of the plant by the crop coefficient provided by Rural Development Administration (RDA) [23].

### 2.4.2. Calculation of Effective Rainfall and Irrigation Water Need

Effective rainfall is determined depending on the difference between total rainfall and actual evapotranspiration. It can be measured directly from the climatic parameters and the usable ground reserves. At ground level, water from effective rainfall is categorized as surface run-off and infiltration. Equation (2) calculates effective rainfall.

$$Re(t) = D(t) - D(T - 1) - Req(t) + ETc(t) \tag{2}$$

where *Re(t)* is the effective rainfall at *t* days (mm), *D(t)* is the soil moisture content at *t* days (mm), *D(t* − 1) is the soil moisture content at *t* − 1 days (mm), *Req(t)* is the net irrigation at 1 day (mm), and *ETc(t)* is the consumptive use (or evapotranspiration) by a crop at *t* dasy (mm).

Equations (3) and (4) show that if the minimum value of *D(t)* is less than the sum of *D(t* − 1) and *Re(t)*, irrigation water is not required. However, if the minimum value of *D(t)* is larger than the sum of *D(t* − 1) and *Re(t)*, then we subtract *ETc(t)*. Irrigation water is then calculated as the sum of the maximum value of *D(t)* and *ETc(t)*, minus the sum of *D(t* − 1) and *Re(t)*.

$$If\ Dmin\ \leq D(t-1) + Re(t), Req(t) = 0 \tag{3}$$

$$If\ Dmin\ \geq D(t-1) + Re(t) - ETc(t), Req(t) = Dmax - D(t\ 1) - Re(t) + ETc(t) \tag{4}$$

### 2.4.3. Measurement of Direct Water Consumption

The amount of irrigation water required is converted to the amount of surface and ground water considering the rate of consumption by the source of water consumed in each region. The converted surface water and ground water usage are changed into direct water consumption by multiplying the water scarcity index by the water source, developed using the water scarcity footprints method proposed by Tokyo University [24,25]. Table 3 shows the water scarcity index applied in this study.

**Table 3.** Water scarcity index by water source.

|  | Precipitation | Surface Water: River | Surface Water: Reservoir | Ground Water |
|---|---|---|---|---|
| Water scarcity index | 1.0 | 2.5 | 6.9 | 35.1 |

## 3. Results and Discussion

### 3.1. Distribution of Weather Data

The distribution of six weather indices was analyzed to estimate the evapotranspiration of each agricultural product (Figure 4). These data were recorded from 79 weather stations in Korea by analyzing the daily weather conditions from 1 January 2003, to 31 December 2012. Approximately 288,000 data points were collected for each weather indicator [11]. Figure 4 shows that all the bar graphs were obtained by plotting the indicator values for the weather condition data on a monthly basis. The highest monthly temperature distribution was the highest in August and the lowest was in January. Additionally, the monthly lowest temperature distribution was lowest in January and highest in July. The duration of sunshine was the longest in July and August (about 14 h). Relative humidity exceeded 50% in the summer of July and August and was less than 20% from January to April. The average wind speed increased to 16 m/s in August and September when typhoons were frequent. June was the lowest. Finally, precipitation was concentrated from July to September.

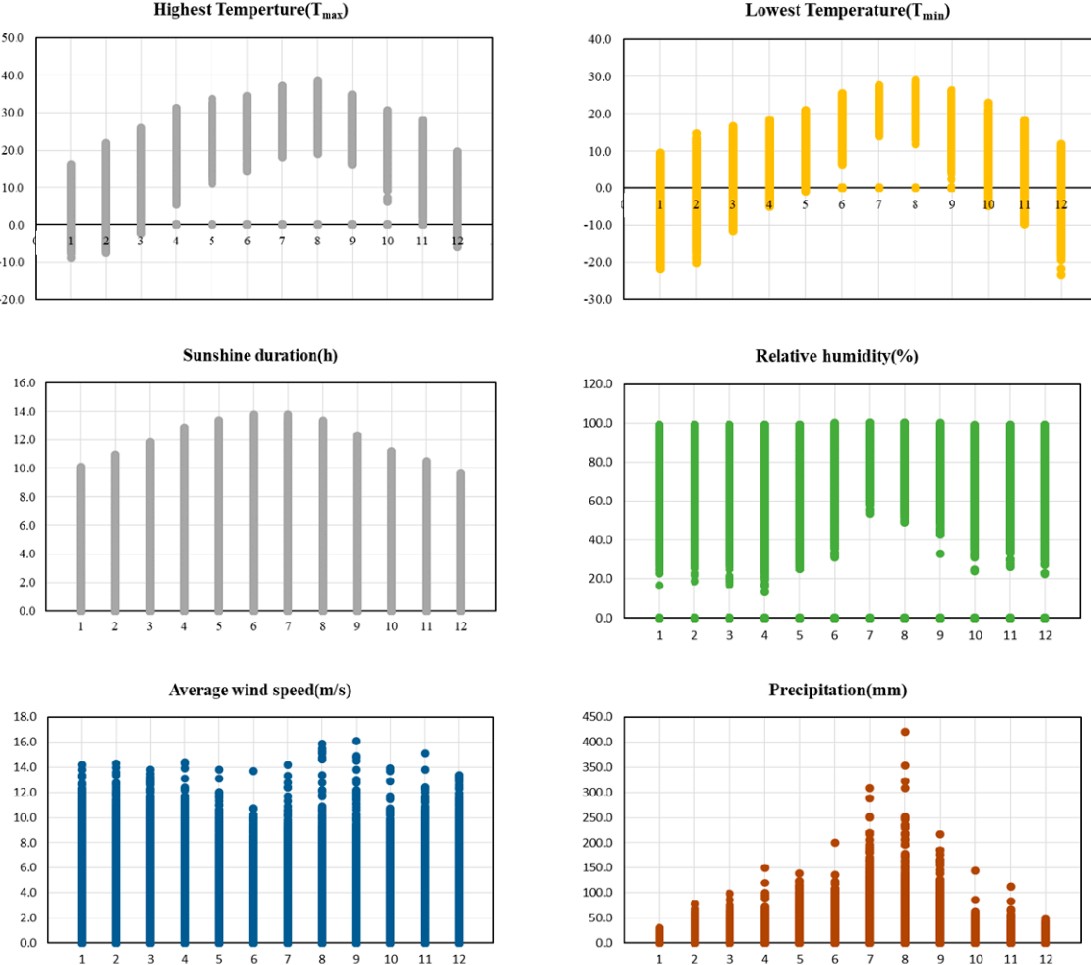

**Figure 4.** Monthly distribution chart of weather data (2003–2012).

Figure 5 presents schematic distribution results of six weather indicators per year. The annual distribution chart appears relatively constant compared to the monthly distribution chart of Figure 4. Maximum temperatures exceeding 35 °C were observed several times every year. The highest temperature in 2012 was estimated to be close to 38 °C and the temperature was the lowest in 2004 and 2007. The duration of sunshine averaged 13.5 h, but, in 2009, it was up to 14 h. The relative humidity

was lowest in 2002, and the average wind speed was highest in 2005 and 2012. Precipitation was the lowest in 2008 and relatively small in 2003 and 2009 compared with other years.

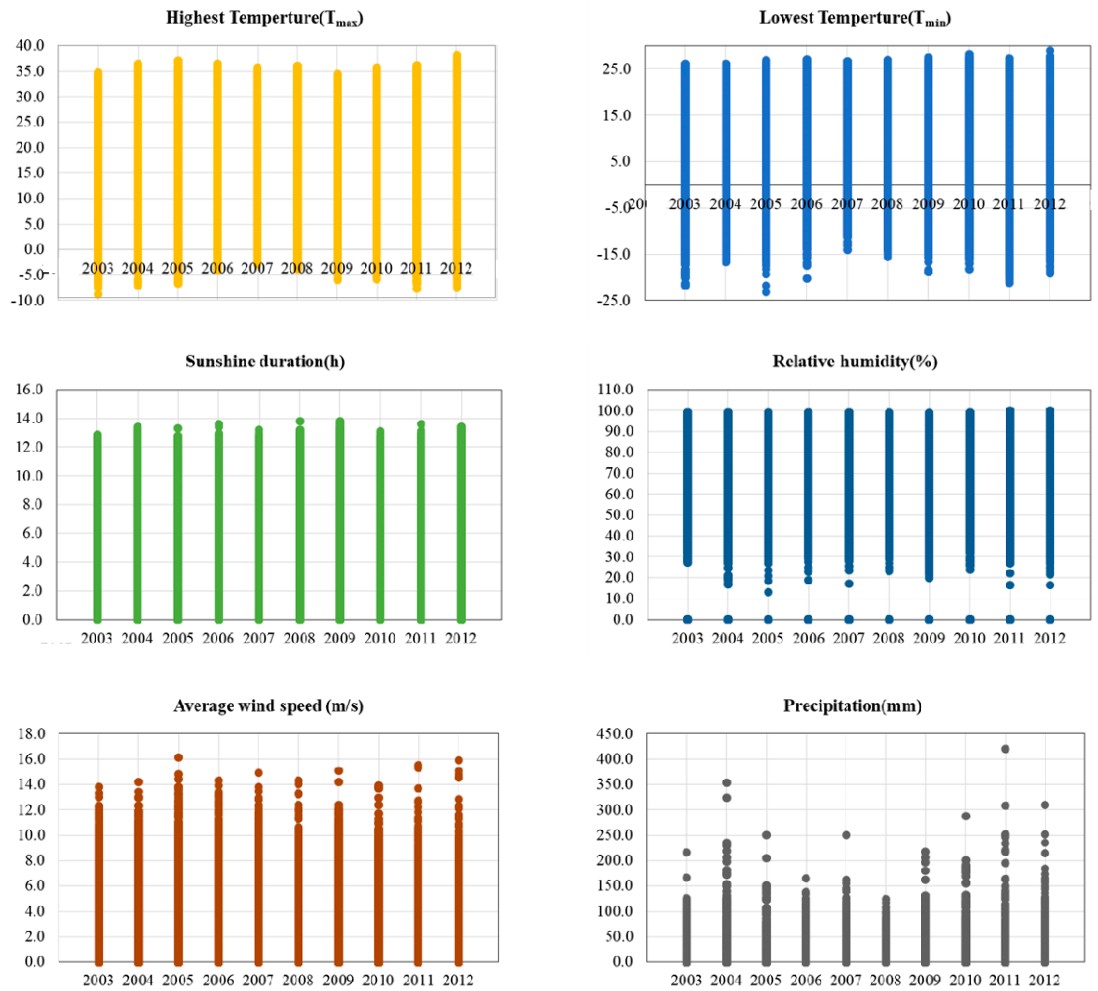

**Figure 5.** Annual distribution chart of weather data (2003–2012).

*3.2. Water Footprint of Agricultural and Livestock Products*

3.2.1. Direct Water Footprint of 43 Agricultural Products on a Hectare Basis

Figure 6 shows the direct water footprint per hectare of 43 agricultural products estimated using the water footprint assessment model developed for this study. Among the products, the water footprint of rice was the largest at 11,741 m³/ha, and that of the autumn potato was the smallest at 2096 m³/ha. In the case of open field, open field vegetables, the footprint of pepper was the highest at 4994 m³/ha, and spinach was the lowest at 2132 m³/ha. With regard to vegetables grown in facilities, the footprint of cucumber was the largest at 34,962 m³/ha and that of spinach was the smallest at 3195 m³/ha. The reason is that the crop coefficient developed considering the growth period of the crop of cucumber is bigger than that of spinach. Among the fruits, the footprint of the grape was the largest at 17,159 m³/ha, and the least was 6813 m³/ha. In total, fruits were considered to have a higher water footprint per unit area than crops and vegetables. The indirect water consumption of vegetables grown in facility was much higher than that of vegetables grown in an open field, because vegetables grown in facility consume more fuel, energy, and water than open field vegetables.

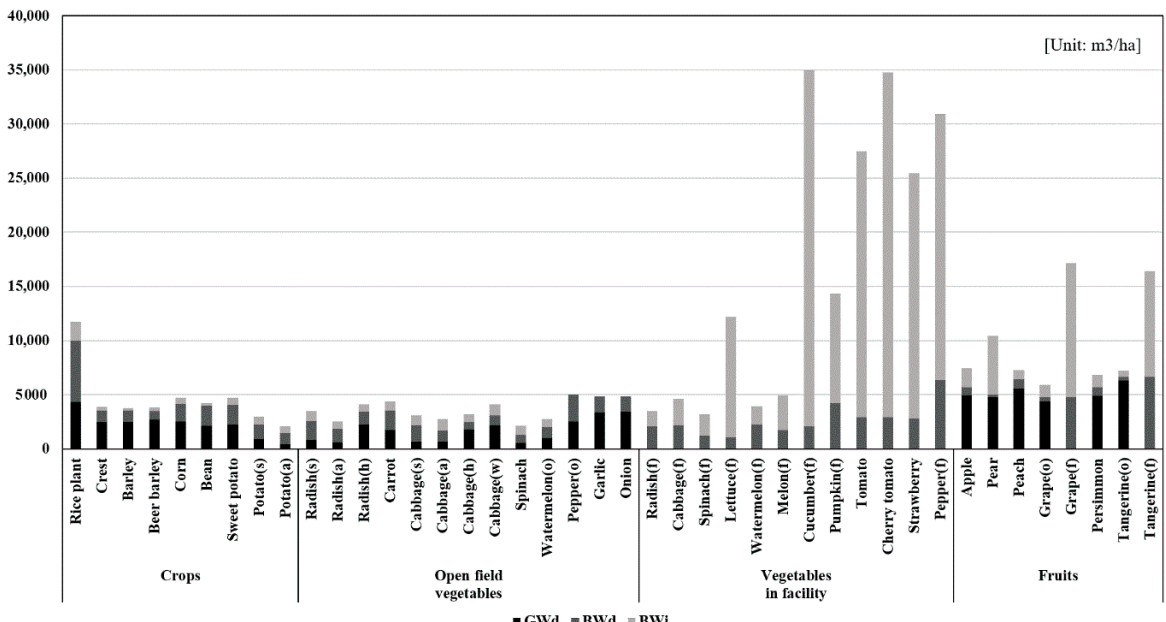

**Figure 6.** Comparison of direct water footprint per ha for 43 agricultural products. Black bar indicates green water as direct water ($GW_d$) and dark gray bar indicates blue water as direct water ($BW_d$), and finally, light gray bar indicates blue water as indirect water ($BW_i$).

### 3.2.2. Direct Water Footprint of 43 Agricultural Products in Ton Basis

Figure 7 illustrates the direct water footprint of 43 agricultural products on a ton basis. As shown in Figure 7, the water footprint of soybean among the crops was the highest at 3859 m³/ton, 2.9 times the average water footprint of crops: 1320 m³/ton. Among the open field, open field vegetables, the water footprint of spinach was the highest at 930 m³/ton, 3.2 times the average of open field vegetables: 287 m³/ton. Next, among the vegetables grown in the facility, the water footprint of strawberries was the largest at 6046 m³/ton: 2.7 times the average of vegetable, 2268 m³/ton. Finally, the water footprint of the grapes cultivated in facility was the highest at 7085 m³/ton: 3.5 times the average of 2027 m³/ton of fruits. Here, the reason why the water footprint was different for each agricultural product is because the crop coefficients were different, as mentioned.

From Figure 7, the average water footprint of the vegetables grown in facility was the highest among the four types of agricultural products, followed by fruits, crops, and vegetables grown in the open field. The average water footprint of vegetables grown in facilities was about 7.9 times the average water footprint of the vegetables grown in the open field.

The pattern of the bar graph shown in Figure 7 is different from the pattern of Figure 6, especially for soybeans, grapes, and cucumbers. For soybeans and grapes, the water footprints per hectare are low, but the water footprint per ton is relatively high. However, the water footprint of cucumber has the opposite pattern compared to those of soybeans and grapes. Thus, yields per hectare of soybean and grape were relatively low compared to other crops, and the cucumber yield per hectare was relatively higher than other agricultural products. Thus, it is not suitable for estimating the water footprint for agricultural products, because the result of water footprint per area did not reflect the production yield per area.

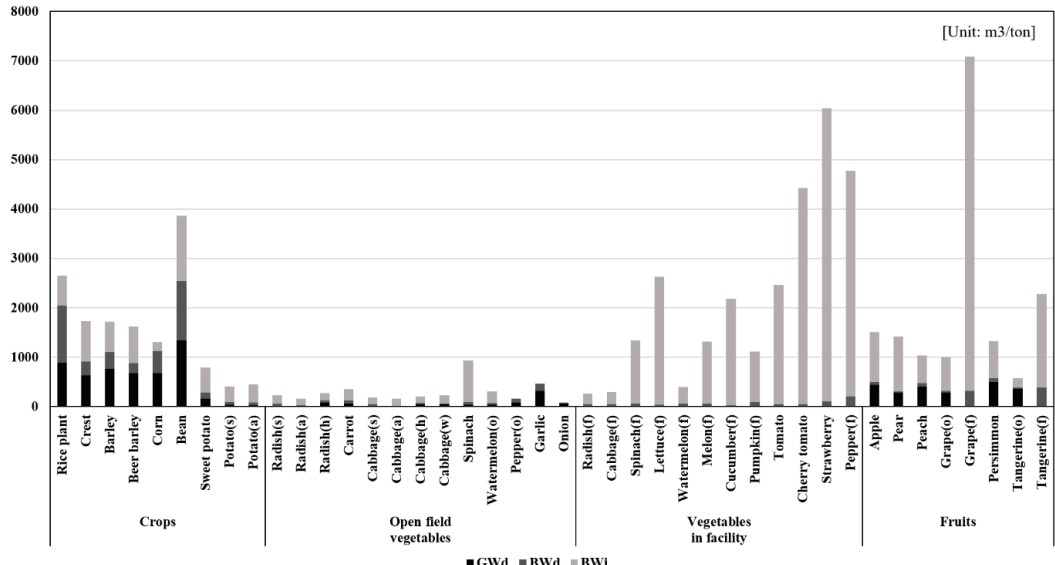

**Figure 7.** Comparison of water footprint per ton for 43 agricultural products. Black bar indicates green water as direct water (GW$_d$) and dark gray bar indicates blue water as direct water (BW$_d$), and finally, light gray bar indicates blue water as indirect water (BW$_i$).

### 3.2.3. Direct Water Footprint of Three Livestock Products on a Ton Basis

Figure 8 illustrates the results of the direct water footprint of three livestock products. The water footprint of beef, including green and blue water from direct and indirect sources, was 19,600 m$^3$/ton. The water footprints of pork and chicken on a ton basis were 5272 m$^3$/ton and 4008 m$^3$/ton, respectively. The water footprint of beef was the largest, because the intake of feed consumed during breeding was higher than that of pigs and chickens. The result of the water footprints of three livestock products show that the water footprint of beef was 3.7 times and 4.9 times pork and chicken, respectively. The blue water footprints of three livestock products were approximately 66%. However, the water footprint per ton of beef was analyzed to be 8.6 times that of the average water footprint per ton of the vegetables grown in facilities. The water footprint per ton of pork and chicken was 2.3 times and 1.8 times larger than that of vegetables grown in facility, respectively. The average water footprint of meat per ton was 4.2 times higher than that of vegetables in facility.

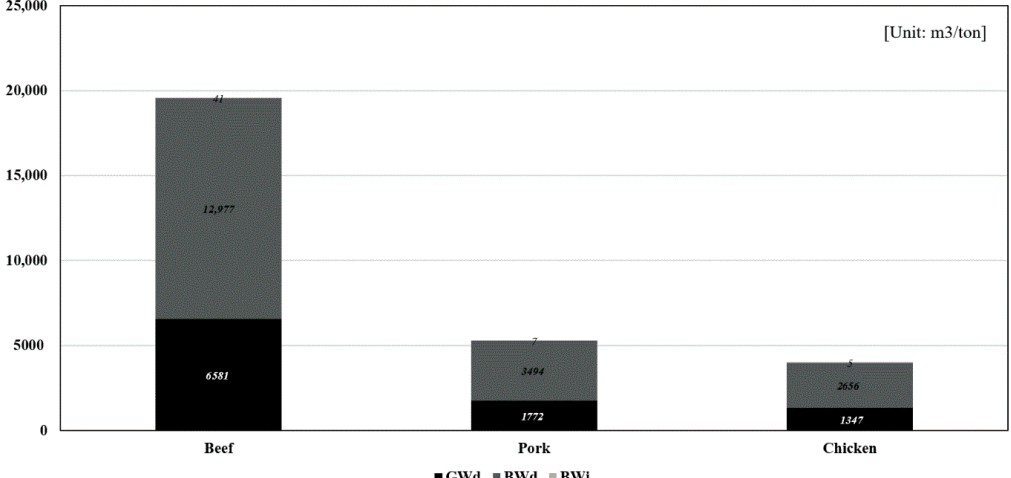

**Figure 8.** Comparison of water footprint per ton for 3 livestock products. Note: In this figure, the black bar means green water as direct water (GW$_d$) and the dark gray bar is blue water as direct water (BW$_d$), and finally, the light gray bar is blue water as indirect water (BW$_i$).

### 3.3. Comparison of Water Consumption per Region

Generally, the water footprint for agricultural products is affected by weather conditions. In Korea, the regional variation of climate in terms of temperature, wind speed, and precipitation is large between the northern and southern regions. Therefore, it is important that water footprints are calculated and compared at the regional level considering their weather characteristics. The subjective product for comparison is rice. The cultivation area for rice spreads nationwide, and the water footprint is larger than other crops.

Figure 9 depicts direct water footprint per region and year. Here, the purpose of analyzing the water footprint only for direct water is that indirect water consumption from agricultural materials such as fertilizer, pesticides, and mulching vinyl is not affected by domestic weather conditions. In 2003, it consumed 9600 $m^3$ of direct water to harvest rice of 1 ha, which is the least consumption in a year. However, the years with the highest direct water consumption were 2004 and 2012, more than 10,300 $m^3$ per 1 ha. The region-wise annual water consumption of Chungcheongnam Province was the largest, and that of Jeollanam Province was the least. Moreover, the central regions, including Gyeonggi Province, Gangwon Province, Chungcheongbuk Province, and Chungcheongnam Province, had higher direct consumption per 1 ha than the southern regions, including Gyeongsangnam Province, Gyeongsangbuk Province, Jeollanam Province, and Jeollabuk Province.

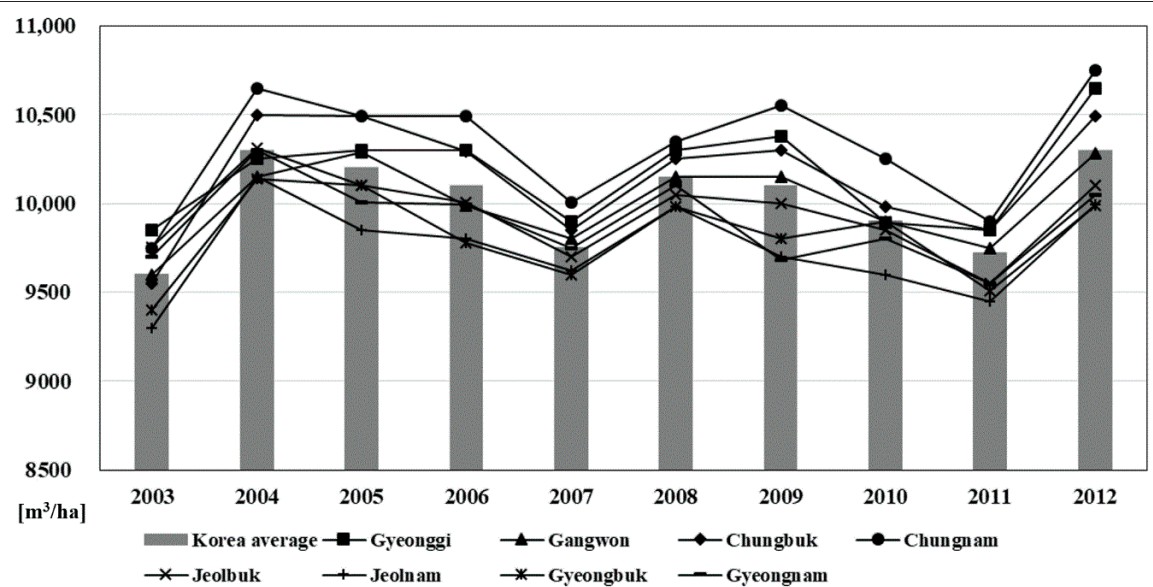

**Figure 9.** Comparison of direct water consumption per ha by region and year.

Figure 10 shows a graph comparing green water consumption by region and year. In 2003, the consumption of green water (6000 $m^3$) was the largest to produce rice at 1 ha. However, green water consumption was lowest in 2004, 2008, and 2009. Whereas there was a difference in precipitation by region per year, Jeollabuk Province had a relatively large consumption of green water. Gyeongsangbuk Province, however, had relatively less. In fact, no consistent trends were seen in the use of green water per year.

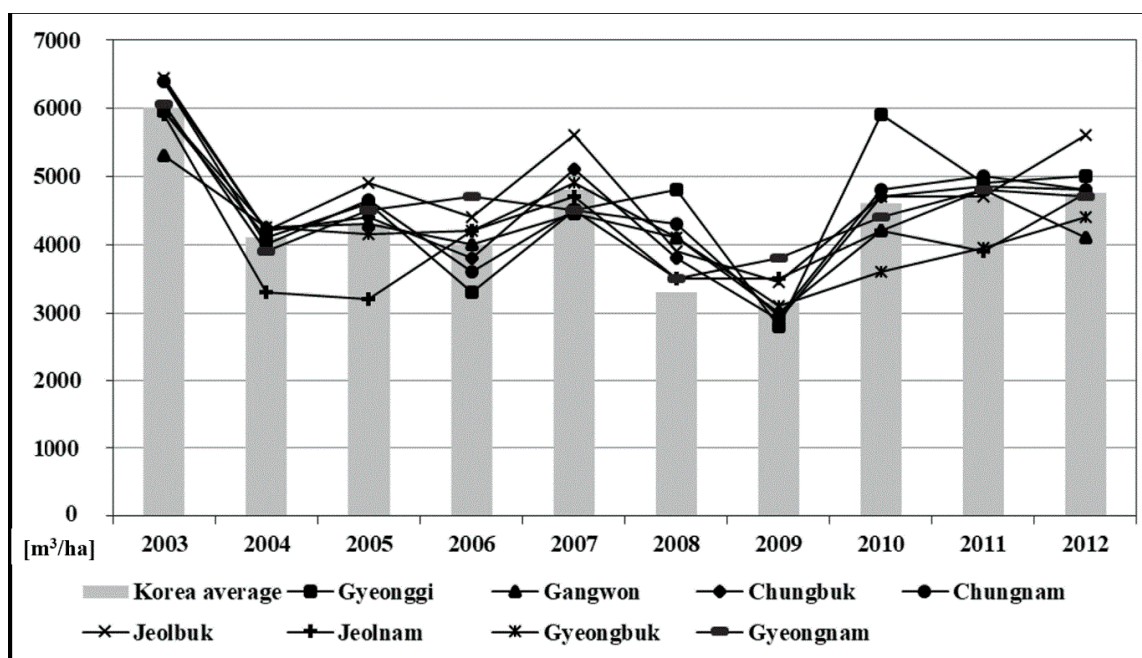

**Figure 10.** Comparison of green water consumption per ha by region and year.

Figure 11 illustrates the amount of blue water consumption by region and year. It shows that the tendency toward blue water consumption by region and year is exactly opposite of the consumption trend of green water, because the consumptive water use per crop, which is same as evapotranspiration, is constant. In fact, Jeollabuk Province had the least amount of blue water usage, whereas Gyeongsangbuk Province had the highest amount.

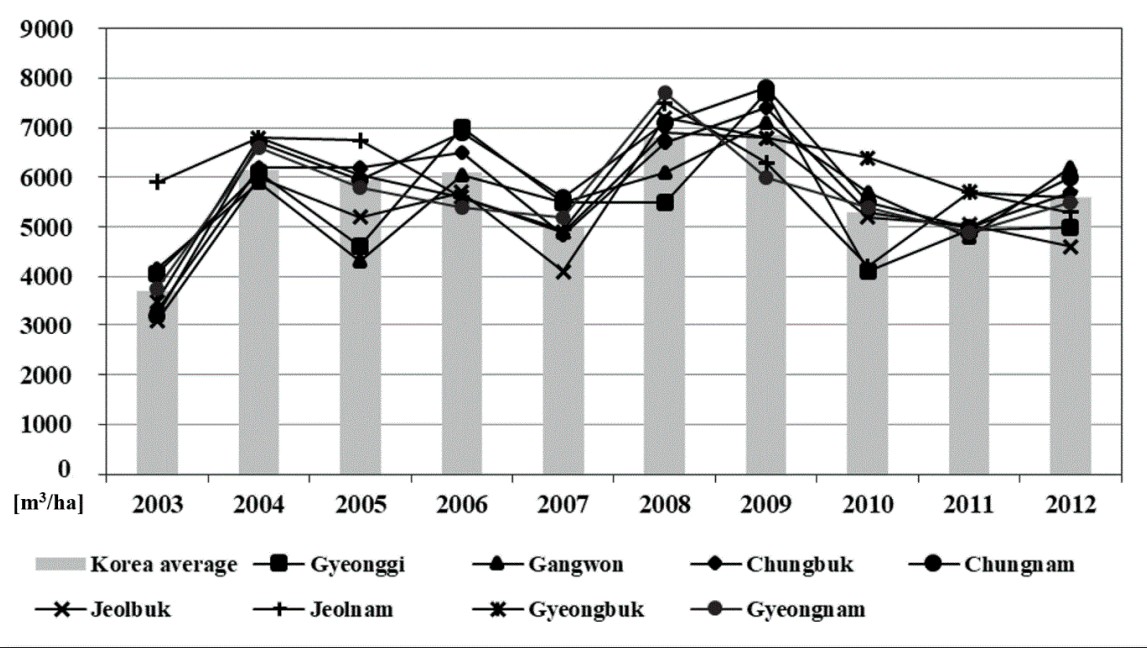

**Figure 11.** Comparison of blue water consumption per ha by region and year.

Figure 12 shows the region- and year-wise distribution of direct water consumption per ton. Comparing the distribution shown in Figure 12 with that of Figure 9, direct water consumption per ton was relatively constant compared with consumption per ha, depending on the year and region.

Consequently, direct water consumption is more closely related to yield than to land area. The average consumption of direct water per ton was 2200 m$^3$, but in 2009, it was 1900 m$^3$ lower than the average consumption. This study investigated why the direct consumption in 2009 was less than other years using Figure 5. In 2009, the sunshine duration was longer, and the average wind speed and relative humidity were the lowest compared to other years. Precipitation was also relatively less than other years. Therefore, the amount of direct water consumption per ton in 2009 decreased, because of increased yield depending on weather conditions.

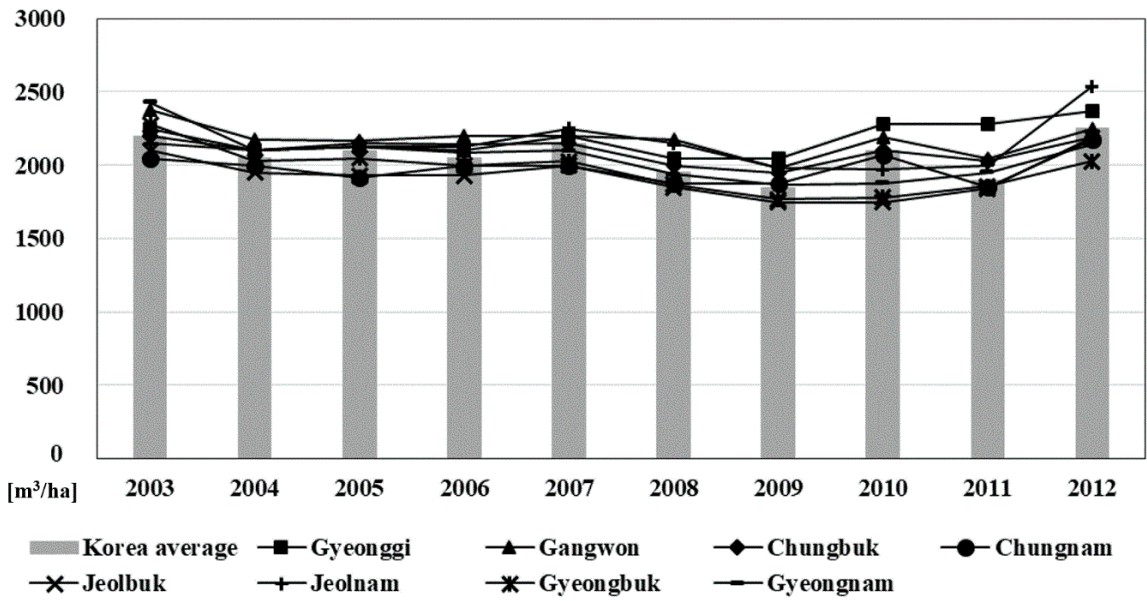

**Figure 12.** Comparison of direct water consumption per ton by region and year.

*3.4. Estimation of Direct Water Footprint of a National Scale in Agricultural Sector*

On the basis of the direct water footprints for 42 agricultural and three livestock products, this study estimated the total water footprint for all agricultural and livestock products consumed in the Republic of Korea in 2014. For the purpose of the study, it was assumed that all agricultural and livestock products consumed were produced domestically. In 2014, the total consumptive amounts of each agricultural and livestock product group were based on data provided by the National Statistical Office.

Figure 13 shows a bar graph representing the direct water footprint required for the production of agricultural and livestock products at the national level. The extrapolation method was used to extend the water footprints per ton of 45 agricultural and livestock products to the water footprint from the total consumption of 45 products. Accordingly, the total water footprint was estimated at 37.6 billion m$^3$, which is equivalent to 27.9% of the total available water resources of Korea, with an average 134.9 billion m$^3$ per year [26]. Next, the total blue water consumption was analyzed to be 69.3% of the national total water footprint. This is equivalent to 19.3% of the total available water resources in Korea. Of the total water footprints, meats accounted for 43.0%, followed by crops and vegetables by 34.1% and 13.3%, respectively. In particular, it was analyzed that the indirect water of vegetables cultivated mainly in facility was close to 90% of the total water footprint of vegetables, and the consumption of blue water by meat accounted for 66.2% of the total water footprint of meat, because the meat consumed mainly processed feed.

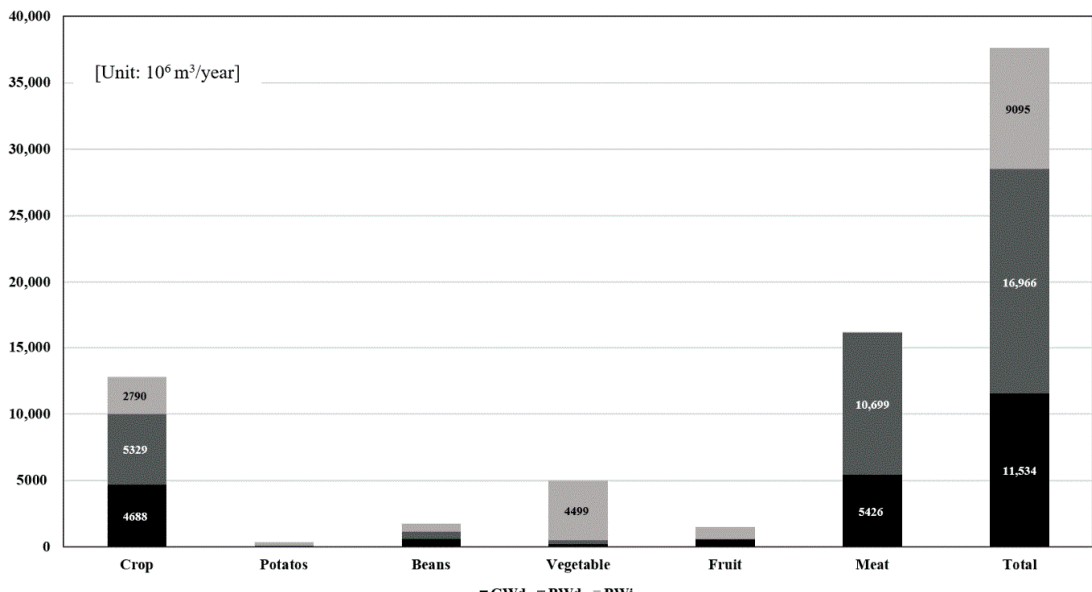

**Figure 13.** Direct water footprint by agricultural and livestock products at a national level.

### 3.5. Comparison of Water Footprints between Different Studies

To verify the reliability of this study, we compared the results of water footprint studies of different rice sources. The comparative sources are Korea Rural Community Corporation and WFN, representative water footprint research institutes and Figure 14 shows a bar graph comparing the water footprint of rice among three different sources [27,28]. According to the results, the direct water footprint as the sum of green, blue, and gray water, excluding indirect water, was almost the same as the three sources. In particular, the results of green water calculated on the basis of weather conditions being in close agreement with the results of WFN. However, the results of blue water were somewhat different from the three studies. This could be caused by differences in the method and timing of collecting statistical data at the farm and open field. However, compared to other studies, the characteristic of this study is that it considers indirect water consumption by agricultural materials according to the requirements of ISO 14044 and ISO 14046. Thus, it concluded that the inclusion of indirect water in the water footprint resulted in about 17% increase.

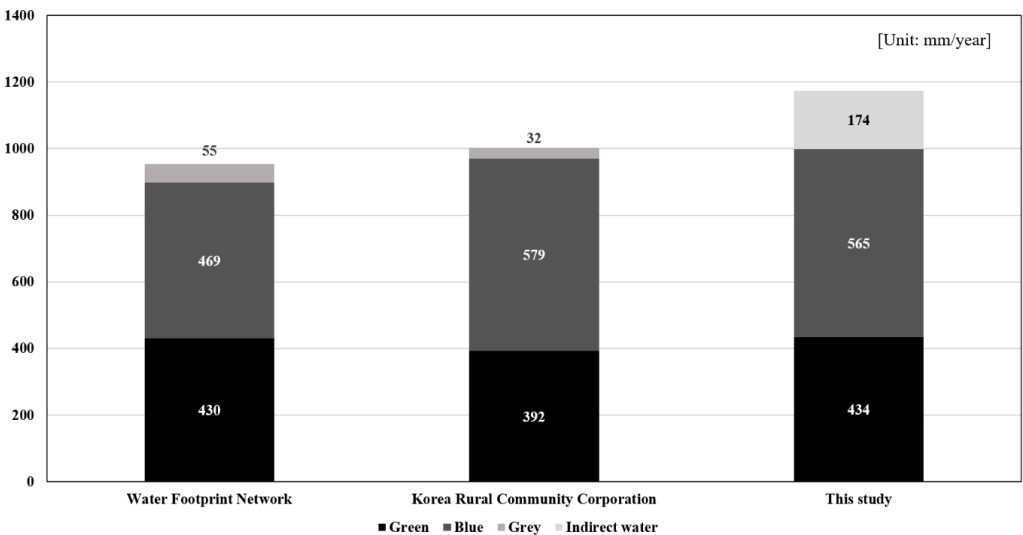

**Figure 14.** Comparison of water footprints on rice at different studies.

## 4. Conclusions

This study measured the water footprints of 42 agricultural products and three livestock products using a method developed by integrating the characteristics of two methods of measuring internationally accepted water footprints. To estimate accurate and representative water footprints, this study collected massive weather data on six indicators from 79 weather stations in Korea from 2003 to 2012. From the results of the water footprint, we confirmed that the average water footprint of the vegetables grown at facilities was 7.9 times larger per ton than those of the vegetables cultivated in an open field. Moreover, we found that the water footprint per ton of beef was about 4.2 times the average water footprint per ton of vegetables grown in facility. Assuming that all agricultural and livestock products consumed in Korea were produced domestically in 2014, the water footprint estimates accounted for 27.9% of the total domestic water resources. This study is meaningful in that it estimated the water footprint of agricultural and livestock products using massive meteorological data measured in Korea. In a future study, we will analyze the amount of the virtual water trade of Korea via the import of food. The results of that study should be effectively applied to overall national agricultural water management, including virtual water trade.

**Author Contributions:** Author 1 (I.K.) wrote the directions and main contents of this paper, and author 2 (K.-s.K.) supported the direction of this paper.

**Funding:** This research was funded by the Institute of Planning and Evaluation for Technology in Food, Agriculture and Forest (IPET) and the Rural Development Administration (RDA).

**Acknowledgments:** This study was carried out with support from the Institute of Planning and Evaluation for Technology in Food, Agriculture and Forest (IPET) and the Rural Development Administration (RDA). We sincerely appreciate the work and support of the IPET and RDA.

**Conflicts of Interest:** The authors declare no conflicts of interest.

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
