# Peer review of "Estimation of Water Footprint for Major Agricultural and Livestock Products in Korea"

_sustainability, doi:10.3390/su11102980_

Reviewer 1 Report

Although the manuscript provides data of WF determination in a national level, that is very interesting to compare with other countries, I regret to say that the overall quality of this work is not suitable yet for publication.

The authors are advised to take in to account all remarks in the manuscript, and reorganize and rewrites from the beginning the manuscript, with the appropriate caution.

Forgotten “copy paste” sections are not permissible in this level of scientific publications.

The “English text” has to be edited by a native English speaker.

The authors are advised however to continue their effort for publication of their data as it seems to be quite interesting.      

Reviewer 2 Report

The paper is interesting and has some relevant merits. Among these, in particular, (1) the integration of the two methodologies (ISO and WFN) and (2) the validaton of the study operated by comparing the results of water footprint on rice from different sources, i.e. the Korea Rural Community Corporation and the WFN. 

The main critics is refererred to the choice of the agricultural products. What about the integration of mealworms in the list of species which are most consumed by the Korean people? Authoprs could find data in (Miglietta et al., 2015). Please try to address this issue, to make the study really complete.

I propose the authors to integrate in the Conclusions section some references about (1) the role of the virtual water trade in relaxing the national exploitation of water resources (Lamastra et al., 2017), (2) the importance of monitoring the groundwater contamination due to agricultural activities and land uses (Serio et al., 2018), the determinants of farmers' intention to adopt water saving measures.

Miglietta, P., De Leo, F., Ruberti, M., & Massari, S. (2015). Mealworms for food: a water footprint perspective. Water7(11), 6190-6203.

Lamastra, L., Miglietta, P. P., Toma, P., De Leo, F., & Massari, S. (2017). Virtual water trade of agri-food products: Evidence from italian-chinese relations. Science of the Total Environment599, 474-482.

Serio, F., Miglietta, P. P., Lamastra, L., Ficocelli, S., Intini, F., De Leo, F., & De Donno, A. (2018). Groundwater nitrate contamination and agricultural land use: A grey water footprint perspective in Southern Apulia Region (Italy). Science of the Total Environment645, 1425-1431.

Author Response

Please find attached file

Round  2

Reviewer 1 Report

Manuscript has been improved in regards to the previous version.

Some basic methodological aspects however have to be clarified. Also in the discussion of the results some more effort has to be given.

Specific parts of the manuscript with relevant comments can be found in the attached file.

Reviewer 2 Report

I did not notice any of the required modifications. In particular the references to the concept highlighted i the first review round. For THE LAST TIME BEFORE REJECTING THE PAPER I ask to implement the required revisions and indicate them clearly in the new version of the Manuscript.

Author Response

Please find attached file

Round  3

Reviewer 1 Report

The manuscript is now significant improved in regards to methodological aspects and results presentation. Some minor improvements suggested can be found in the attached file.

English language however can be improved significantly and i strongly reccoment a detailed editing of English language and style by native english speaker before publication.

Author Response

Dear Reviewer

I revised my submission as follows;

I didn't insert your comment such as average irrigation and yield on agricultural products in this submission. Because, irrigation was estimate value from Penmann equation, not collective data. Also I would like to add your comment in the next paper. 

I reviewed English text of my submission by native English speaker.

Reviewer 2 Report

The paper can be published after a moderate English editing.

Author Response

Dear Reviewer

I reviewed English text of my submission by native English speaker.